# Features Found in Indocyanine Green-Based Fluorescence Optical Imaging of Inflammatory Diseases of the Hands

**DOI:** 10.3390/diagnostics12081775

**Published:** 2022-07-22

**Authors:** Egbert Gedat, Jörn Berger, Denise Kiesel, Vieri Failli, Andreas Briel, Pia Welker

**Affiliations:** 1Faculty of Engineering and Natural Sciences, Wildau Technical University of Applied Sciences, 15745 Wildau, Germany; 2Xiralite GmbH, 10115 Berlin, Germany; berger@xiralite.com (J.B.); failli@xiralite.com (V.F.); briel@xiralite.com (A.B.); 3Institute of Functional Anatomy, Charité Universitätsmedizin Berlin, 10115 Berlin, Germany; denise.kiesel@charite.de (D.K.); pia.welker@charite.de (P.W.)

**Keywords:** optical diagnosis imaging, fluorescence optical imaging, indocyanine green fluorescence, image feature, image analysis, rheumatoid diseases

## Abstract

Rheumatologists in Europe and the USA increasingly rely on fluorescence optical imaging (FOI, Xiralite) for the diagnosis of inflammatory diseases. Those include rheumatoid arthritis, psoriatic arthritis, and osteoarthritis, among others. Indocyanine green (ICG)-based FOI allows visualization of impaired microcirculation caused by inflammation in both hands in one examination. Thousands of patients are now documented and most literature focuses on inflammatory arthritides, which affect synovial joints and their related structures, making it a powerful tool in the diagnostic process of early undifferentiated arthritis and rheumatoid arthritis. However, it has become gradually clear that this technique has the potential to go even further than that. FOI allows visualization of other types of tissues. This means that FOI can also support the diagnostic process of vasculopathies, myositis, collagenoses, and other connective tissue diseases. This work summarizes the most prominent imaging features found in FOI examinations of inflammatory diseases, outlines the underlying anatomical structures, and introduces a nomenclature for the features and, thus, supports the idea that this tool is a useful part of the imaging repertoire in rheumatology clinical practice, particularly where other imaging methods are not easily available.

## 1. Introduction

In recent years, fluorescence optical imaging (FOI) has emerged as a valuable tool for the diagnosis of inflammatory diseases of the hands. Inflammation alters the microvascularization and permeability of the vessels, which, in most cases, leads to an increased accumulation of the contrast agent ICG in the affected regions. Visualization of those phenomena has proved useful in supporting the diagnosis of numerous joint inflammation diseases, such as rheumatoid arthritis [1,2], osteoarthritis [3,4], connective tissue diseases [5,6], psoriasis arthritis [7,8,9], and others [10,11]. Those results were recently summarized in a review by Ohrndorf et al. [12]. Most studies analyze signal intensity in the region of the finger joints, i.e., the distal interphalangeal joints (DIP), proximal interphalangeal joints (PIP), metacarpophalangeal joints (MCP), and the carpus for the diagnostic assessment. Only a few authors use different image features for the diagnosis of connective tissue diseases [5,13] and psoriasis arthritis [8,14,15,16,17]. However, other anatomical structures can also be affected by inflammatory disorders, including, but not limited to, entheses, tendons, muscles, and skin. Interestingly, FOI allows detection of microcirculatory changes in those structures as well. However, knowing from which anatomical structure the signal is coming from and using it in the diagnostic process requires an additional consideration. Current technology cannot distinguish from which depth (and therefore structure) the signal is coming. Nevertheless, this information can be extracted by carefully evaluating when and how FOI signal is detected. Therefore, this work will summarize the important image features (location, shape, and timing) found in patients with various rheumatic diseases. Patterns of enhancement of a contrast agent in the FOI examinations of patients are characterized and assigned to inflammatory anatomical structures, which are known to be typical for the diseases in clinical diagnosis. The diagnosis of rheumatic diseases is often difficult due to the overlapping of symptoms and affected structures. We have focused here on attributing the patterns of the FOI examinations to the anatomical structures and not discussing a diagnosis. Thus, a basis is set for discussion and for analyses of larger numbers of patients in later studies and to carry out statistical analyses leading to prove a possible diagnostic value of the features.

## 2. Materials and Methods

A database of 3690 patients with FOI examinations was available, which was a collection of data from several clinical partners with a specialty in rheumatology. The patients were diagnosed with: suspicion of early arthritis (291); rheumatoid arthritis (1174); psoriasis and suspicion of psoriatic arthritis (741); connective tissue diseases (148); and the remaining were diagnosed with metabolic arthropathies, spondylarthritis, and infectious arthritides. Of the patients, 111 had multiple examinations, with up to 6 follow-up examinations. For each patient, clinical and laboratory parameters, as well as the diagnoses established, were included in the database. Clinical parameters included: age, sex, medical history, ongoing treatments, and a summary of X-ray analysis. Laboratory parameters included: rheumatoid factor status and C-reactive protein (CRP) levels. All FOI data were visually inspected. After thorough analysis, several patients of various disease groups were selected as typical cases. Inclusion criteria were the presence of typical image features and the absence of possible image artefacts. The latter may occur due to delayed or premature injection of the contrast agent, as well as artificial fingernails or tattoos. Hence, eight cases were chosen to be displayed: a 51 y/o male healthy volunteer; a 45 y/o male patient with clinically diagnosed (c.d.), i.e., before inspection of FOI data, rheumatoid arthritis; a 55 y/o female patient with c.d. osteoarthritis; a 55 y/o female patient with c.d. fibromyalgia; a 58 y/o female patient with c.d. CREST (calcinosis, Raynaud’s phenomenon, oesophageal dysmotility, sclerodactyly, and telangiectasia); a 68 y/o male patient with c.d. acrodermatitis chronica atrophicans; a 37 y/o female patient with c.d. psoriasis arthritis; and a 45 y/o male patient with c.d. psoriasis vulgaris. FOI examinations were conducted using the X4 Xiralite^®^ NIR-Fluorescence imaging system (Xiralite GmbH, Berlin, Germany) following the protocol given by Werner et al. [14]. After injection of 0.1 mg/kg/body weight of ICG, NIR images (at excitation wavelength λ < 760 nm and emission wavelength λ > 800 nm) were recorded every second over a period of 6 min. The intensity of the dye concentration is displayed via a false-color grading; white represents a very high concentration and intensity, followed by red, yellow, and green in descending order. To ensure that no information about the signal strength is lost due to saturation, each individual image is set to its maximum (dynamic mode). The resulting 360 images were visually analyzed to highlight “distinctive” image features. Since the contrast agent dynamics vary over time depending on the underlying anatomical and pathological situations, the cluster of 360 images was subdivided in three distinct phases. Phase 1 (P1) corresponds to an early influx phase, phase 2 (P2) is a mid-phase of distribution of the contrast agent, and phase 3 (P3) represents the late flow after phase 2. Phase images were generated by summing up all images of a phase. Finally, a stack of images, 1–240, is summed-up in a fused overlay image called Prima Vista Image (PVI). This allows the visualization of the most prominent features in all joints for both hands and wrists (30 joints in total) in one single picture. The respective images used for the summations in each patient are given in the images. For each patient, image features are described and if possible, assigned to an anatomical structure. To help us in the analysis, an alphabet letter was assigned to each specific signal pattern. The movies are available in raw and standardized (mp4) format [18].

## 3. Results

### 3.1. Healthy Individual—Reference

To facilitate the interpretation of the FOI signal intensity in patients with various rheumatic diseases, first typical images seen in a healthy volunteer without clinical signs of inflammation are displayed. The subject is a 51 y/o volunteer. As seen in Figure 1, there are no detectable inflammatory or degenerative changes in the hands. Signal increase can only be seen in the very superficial capillary network of the nail bed (f) and a slight signal enhancement in the muscle–tendon junctions (Y) in the late phase. When the signal maximum is reached in the fingertips (Figure 1, left hand), there is a physiological time delay in the right hand depending on the application site of the contrast medium, which must be considered when evaluating pathological findings. No further focal contrast enhancement is observed, denoting absence of articular inflammation.

### 3.2. Rheumatoid Arthritis

The second subject is a 45 y/o male patient with clinical diagnosis of rheumatoid arthritis (Figure 2). The subject tested positive for rheumatoid factor, CRP 18.5 mg/L, but with no signs of erosion in the joints of the hands (X-ray analysis). Accumulation of ICG in P1 is particularly noticeable in the extensor carpi ulnaris (ECU, e), the well capillarized and superficial synovial membrane of the joints, preferably in MCP (M) and PIP (P). In later phases, this effect is not so clearly visible. The composite image (PVI) also shows a signal increase in the carpal articulation/ECU region (C, e) and accompanying venous vessels (V), draining from the affected joints (MCP, PIP).

### 3.3. Osteoarthritis (with Tendinitis)

The selected subject is a 55 y/o female patient with clinical diagnosis of osteoarthritis and tendinitis. The individual is rheumatoid factor negative, CRP < 5 mg/L, with no signs of erosion in the joints of the hands (X-ray analysis). In such a condition, degeneration mainly affects less vascularized connective tissue structures, such as tendons and cartilage, therefore only a small signal increase is detected in the early P1 phase (not shown). In contrast, in the later P2 and P3 phases (Figure 3), there is a significant signal increase in the tissue of tendons (B) and entheses (E). Very typical is also a strong signal around the joints but not on the inner synovial layer, indicating an inflammation most likely in the aponeurosis, or fibrous membrane (O), the outer structure of the joint capsule. The very late accumulation of dye in the transition area from the tendons to the deeper muscles in the forearm is particularly noticeable (Y).

### 3.4. Fibromyalgia

Patients diagnosed with fibromyalgia are difficult to diagnose and, by definition, have no inflammatory tissues. Interestingly, numerous patients that were diagnosed with primary fibromyalgia using conventional clinical examination methods displayed some degree of inflammation in the hands with the FOI examination. We therefore believe that those patients, or at least a subpopulation, might have subclinical inflammatory processes in connective tissue and muscles. The selected subject is a 59 y/o female patient with clinical diagnosis of fibromyalgia, which was tested rheumatoid factor negative, CRP < 5 mg/L, with no signs of erosion in the joints of the hands (X-ray analysis). As seen in Figure 4, signal enhancement in muscle–tendon junctions (Y) is detected in later stages of the examination (P3). Connective tissue structures with less blood supply, such as tendons (B), deeper muscular structures (m), or entheses I are visible as well. A disturbed blood flow to the fingers leads to irregular nail beds (I), which are visible in all phases of the examination. In the earlier phases of the study there is only marginal accumulation of the dye, which is demonstrated in the PVI image as a composite image of all phases.

### 3.5. Collagenosis-CREST Syndrome

CREST (Calcinosis, Raynaud’s phenomenon, oesophageal dysmotility, sclerodactyly, and telangiectasia) syndrome belongs to the group of collagenoses (connective tissue diseases without the involvement of rheumatoid arthritis) which includes inflammation in the blood vessels (vasculitis). The selected subject is a 59 y/o female patient with clinical diagnosis of CREST, tested rheumatoid factor negative, CRP 5.1 mg/L, with no signs of erosion in the joints of the hands (X-ray analysis). As seen in Figure 5, what is typically visible with this condition are Raynaud’s phenomenon (R) and delayed inflow in the fingertips (r). In P1 the punctual signal (F) can be explained by superficial inflammations/defects in the skin while the cloudy signal (W) could be caused by changes in the deeper vessels. Some areas show almost no fluorescence signals (sclerosis, calcinosis) in all phases.

### 3.6. Acrodermatitis Chronica Atrophicans

Acrodermatitis chronica atrophicans (ACA) is the third/late stage of Lyme borreliosis, a condition caused by the Borrelia bacterium. ACA takes a chronically progressive course and finally leads to a widespread atrophy of the skin. Inflammation of the joints can also occur as the disease progresses. The selected subject is a 68 y/o male patient with the clinical diagnosis Lyme borreliosis. The patient had no arthralgia, but swelling and redness on the back of the left hand could be diagnosed on clinical examination. Further diagnostics revealed: rheumatoid factor negative, CRP 61.2 mg/L, and no signs of erosion (X-ray analysis).

As seen in Figure 6, there is an irregular flow (I) of the dye into the fingertips. The superficial inflammation of the skin, which is associated with increased perfusion and swelling, is clearly shown as a punctual (F) and cloudy (W) signal increase in P1 during the early P1 phase. The same area has a strong signal increase during the mid-phase P2 with slight accumulation of the dye in the entheses and tendons €. In the area of the joints, no significant signal changes can be determined in this patient. Results are compatible with diagnosis of Acrodermatitis Chronica Atrophicans (ACA) in the context of Lyme disease.

### 3.7. Psoriatic Arthritis and Psoriasis Vulgaris

Another dermatological disease with potential joint involvement is psoriasis vulgaris which frequently occurs in association with skin and nail psoriasis. Some patients develop psoriatic arthritis as part of the disease. This process typically involves inflammation of the joints in fingers and the surrounding connective tissue. All joints of a finger are often affected (dactylitis). Two distinct cases are shown. The first subject is a 37 y/o female patient with clinical diagnosis of psoriasis arthritis, tested rheumatoid factor negative, CRP 18 mg/L, with no signs of erosion (X-ray analysis). In the left panel of Figure 7, a segmental enrichment of the fluorescence dye in DIP (D), PIP (P), MCP (M) with sigmoid patterns (a) of signal enhancement from the nail bed to the DIP joint can be seen. It is interesting to note that the sigmoid patterns tend to be more visible in P2 and P3. The second subject is a 45 y/o male patient with a clinical diagnosis of psoriasis vulgaris, tested rheumatoid factor negative, CRP 0.8 mg/L, with no signs of erosion (X-ray analysis). Here, in the right panel of Figure 7 skin inflammation in psoriasis plaques (yellow arrows), as well as decreased signal in keratinized plaque (red arrow) are visible.

### 3.8. Summary of Selected Features

In the FOI images displayed above, several distinguishable features of fluorescence signaling are present. They were collected in Table 1. To improve clarity and facilitate future image analysis, features were regrouped according to their localization (e.g., fingertips). Each feature received a nomenclature (the alphabet letters already displayed in Figure 1, Figure 2, Figure 3, Figure 4, Figure 5, Figure 6 and Figure 7) together with a clear description of the signal (e.g., round or oval) and, finally, in which of the figures (selected conditions) they can be found.

## 4. Discussion

Until now, most studies evaluating inflammatory diseases with the support of fluorescence optical imaging have been focused on inflammatory arthritides. In most cases, diagnosis is established with the support of a single picture stack called PVI. This overlay fusion of 240 pictures has the advantage of being automatically generated by the device and still displays changes in the microcirculation of both hands simultaneously. In many cases, PVI is sufficient to establish an accurate diagnosis, with the support, of course, of clinical and laboratory parameters. However, this approach has several limitations. First, signal alteration (enhancement or reduction), which happens for a brief period, might diminish or even disappear once all pictures are stacked together (dilution of signal). Second, and more importantly, the dynamics of dye distribution are totally overlooked, further complicating the task of understanding the specific anatomical localization of the inflammatory processes. Therefore, three aspects must be considered: (1) the time course of the signal variation is influenced by the position in relation to the blood flow (arterial/capillary/venous); (2) the signal strength depends on the degree of perfusion; (3) the depth of the structure determines how much fluorescence signal is attenuated by the overlying tissue. In addition, inflammatory processes change the permeability of the endothelium, inducing an extravasation of the dye.

Although time kinetics have already been considered by some groups (classification into three phases was usual), other anatomical aspects have so far received little attention [3,4,12]. Table 1 collects the signal features that were observed on patients that were analyzed so far. Considering the aspects described above, those patterns can be assigned to the most likely anatomical and pathologically altered structures, considering the aspects described above. Table 2 summarizes this assignment of features to structures, including the time course of the signals. Some of the features described and summarized in Table 2 have already been reported by other authors for the corresponding diagnoses [4,5,6,8,10,14]. Other patterns of enrichment are documented here for the first time. To our knowledge, this study is the first attempt at correlating the exact anatomical structure (tissue) in which disease-related microcirculatory changes have occurred. Based on our observations, but mostly on existing literature, it appears clear that certain features are very likely to be found in certain conditions more than others. For example, patients diagnosed with rheumatoid arthritis have a 50% chance of developing ECU tenosynovitis [19] and a 60% probability of having a synovitis in the MCP joints [20], while these features are extremely rare in patients with degenerative osteoarthritis.

Although it appears clear that some features are characteristic of certain conditions, most will be found in others as well. For instance, inflammatory reactions in the forearm can also occur in otherwise healthy patients as a result of degenerative processes associated with advancing age, trauma, or overexertion in the context of professional activities (Figure 1). These degenerative changes, which extend from the forearm muscles to the power-transmitting tendons are found in many patients and may often be the cause of diagnostics by rheumatologists. The very late signal increase in P3 is typical of the enrichment pattern.

The present work does not state any diagnostic value of the demonstrated features, only their appearance in some typical cases is shown to draw attention to their possible value in the diagnostic process. However, other groups have already compared the method with other imaging techniques, including statistical evaluation [1,2,3,4,7,11]. There are comparisons with X-ray, ultrasound, and MRI scans. Moreover, some work on differential diagnosis aided by FOI has been performed. First results were published for connective tissue diseases [21] and rheumatoid arthritis, osteoarthritis, and psoriatic arthritis [22]. Limitations of the study are, first, that only a small number of examinations can be shown as typical cases, and the choice of these cases is subjective. However, due to the large number of inspected examinations, the chosen cases are the result of well-trained knowledge. Second, “only” 18 image features in FOI examinations are presented here. There are certainly more image features that may be relevant for diagnosis, maybe even features that are hard to see for the human eye, but that can be detected with the aid of computers. They may be introduced, discussed, and added in future work. Third, there is surely a considerable amount of variability in the image features of the patients, also with identical clinical diagnosis. This cannot be addressed here but will be an interesting topic for upcoming analyses.

In future, such analyses will be performed to test large groups of patients for several diseases. These studies will aim at establishing the likelihood of a feature to be present in and specific to certain conditions. Artificial intelligence could prove a useful tool in such analysis and could lead to the development of an algorithm that could support more precise differential diagnosis of rheumatic diseases, as well as the selection of an effective therapy and monitoring.

## 5. Conclusions

The present work displays 18 typical image features found in eight carefully selected subjects, seven with different inflammatory diseases, and one healthy volunteer. To facilitate discussion and future analyses, a nomenclature for those features was proposed. For 16 features, a probable anatomical location was proposed, as well as a temporal appearance in FOI examinations and expected changes in tissue perfusion. The given image examples relating to the patients’ status is expected to advance the diagnostic process with FOI examinations. Other work, as well as future work of the authors, addresses the statistical relevance of the features in the diagnostic process.

## Figures and Tables

**Figure 1 diagnostics-12-01775-f001:**
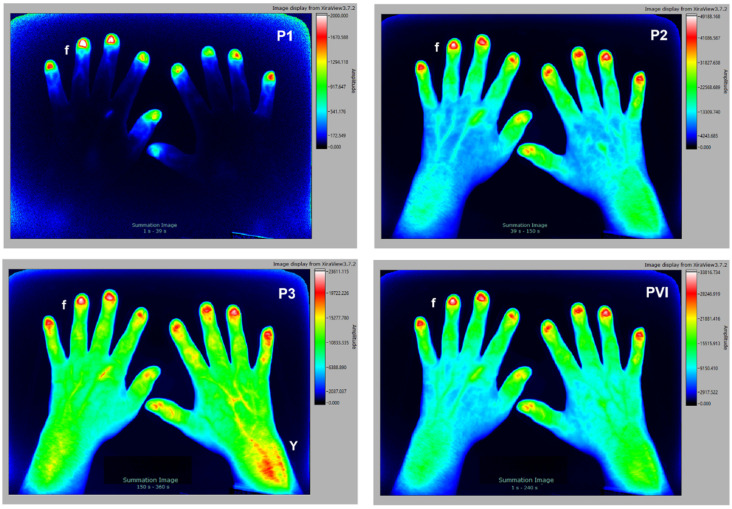
The 51 y/o male volunteer without clinical signs of inflammation. Signal enhancement is seen in the fingertips (f) and a slight signal enhancement in the muscle–tendon junctions (Y).

**Figure 2 diagnostics-12-01775-f002:**
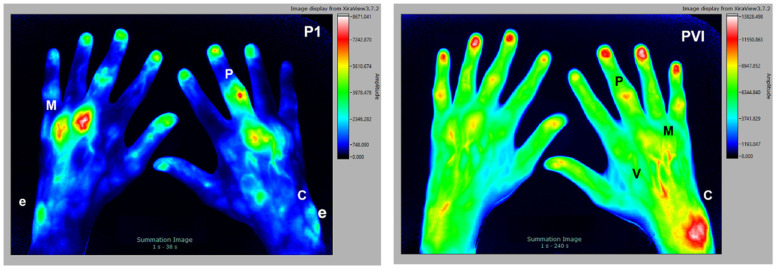
The 45 y/o male patient with clinical diagnosis of rheumatoid arthritis. Signal enhancement seen in MCP (M), PIP (P), carpal articulation region (C), extensor carpi ulnaris (e) region, and venous vessels (V).

**Figure 3 diagnostics-12-01775-f003:**
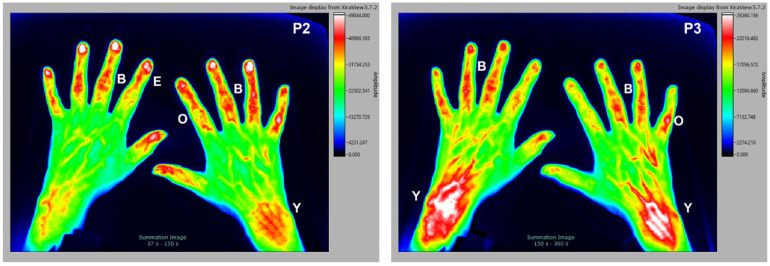
The 55 y/o female patient with clinical diagnosis of osteoarthritis. Signal enhancement seen around the joint capsules (O), in the connective tissue of the entheses (E) and tendons (B) of the fingers with a maximum in P2. Signal enhancement of the muscle–tendon junctions (Y) is visible in P3.

**Figure 4 diagnostics-12-01775-f004:**
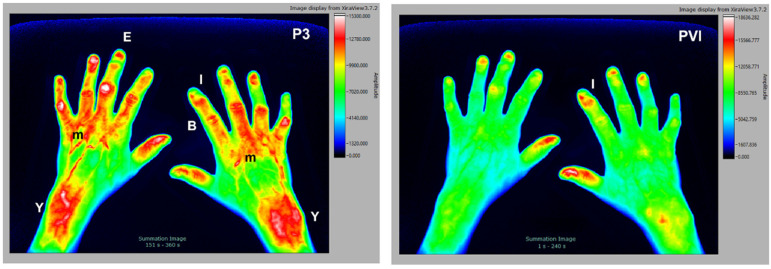
The 59 y/o female patient with clinical diagnosis of fibromyalgia. Signal enhancement seen in muscle–tendon junctions (Y), tendons (B), deeper muscular structures (m), and entheses (E) with irregular nail beds (I).

**Figure 5 diagnostics-12-01775-f005:**
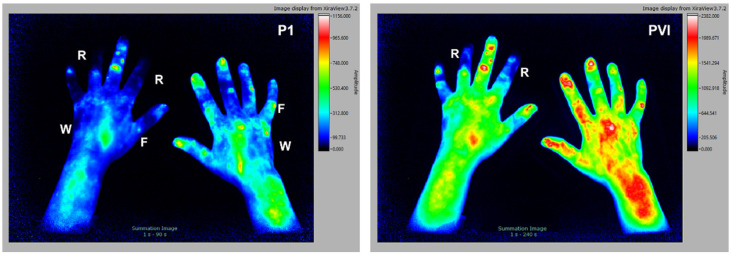
The 58 y/o female patient with clinical diagnosis of CREST. Signal alteration include Raynaud’s phenomenon (R), delayed inflow in all fingertips of both hands, punctual (F), and cloudy (W) signals in P1.

**Figure 6 diagnostics-12-01775-f006:**
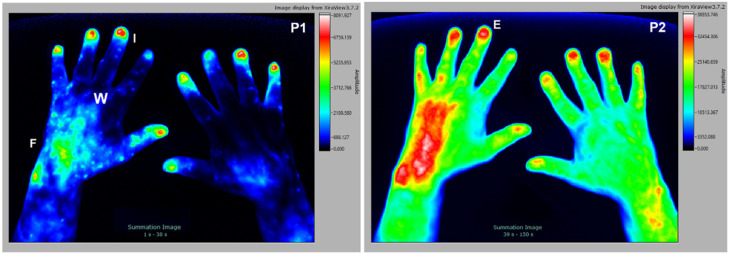
The 68 y/o male patient with clinical diagnosis Lyme borreliosis. Punctual (F) and cloudy (W) signals and irregular dye flow in the fingertips (I) can be seen during P1, with slight accumulation of the dye in the entheses and tendons (E), seen during P2.

**Figure 7 diagnostics-12-01775-f007:**
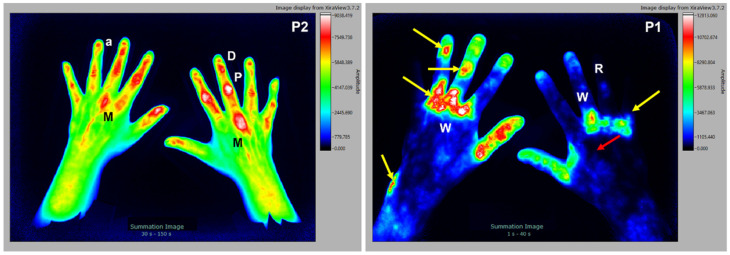
Left panel: the 37 y/o female patient with psoriasis arthritis. Signal enrichment is seen in DIP (D), PIP (P), MCP (M), and sigmoid patterns (a) below the nail bed. Right panel: the 45 y/o male patient with psoriasis vulgaris. Cloudy signal enrichment (W) is seen at the back of the hands and forearms, and in the psoriasis plaques (yellow arrows) with decreased signal in keratinized plaque (red arrow), as well as Raynaud’s syndrome (R).

**Table 1 diagnostics-12-01775-t001:** Selected features seen in FOI images.

Region	Name	Description	Figure
Fingers	f	Round or slightly oval signal in fingertips, physiological	Figure 1
	r	Delayed inflow in all fingertips (in both hands)	Figure 5
	R	Diminished or non-existent flow of the dye into one or more fingers (Raynaud syndrome)	Figure 5 and Figure 7
Nails	a	Sigmoid outflow of the dye from the nailbed	Figure 7
	I	Irregular (inhomogeneous) signal in the nail bed	Figure 4 and Figure 6
Joints	D	Round, oval, mostly regularly shaped signal (DIP joints)	Figure 7
	P	Round, oval, mostly regularly shaped (PIP joints)	Figure 2 and Figure 7
	M	Round, oval, o mostly regularly shaped (MCP joints)	Figure 2 and Figure 7
	C	Round, oval, mostly regularly shaped (Intercarpal joints)	Figure 2
	O	Signal around joints, most likely the fibrous membrane, or aponeurosis	Figure 3
Venous vessels	V	Signal on the back of the hand, in the area of superficial venous structures	Figure 2
Connectivetissue	E	Triangle shaped signal enhancement below nail	Figure 3, Figure 4 and Figure 6
	B	Broad, pronounced signals in the area of dorsal tendons	Figure 3 and Figure 4
	Y	Increased signals in the area of muscle tendon junction of wrist	Figure 1, Figure 3 and Figure 4
	e	Increased signals in the extensor carpi ulnaris region	Figure 2
Muscle	m	Varying intensity and signal sharpness depending on the depth	Figure 4
Skin, connective tissue with blood vessels	W	Cloudy, “unsharp” signal	Figure 5, Figure 6 and Figure 7
	F	Punctual, “sharp” signal, or irregularly shaped	Figure 5 and Figure 6

**Table 2 diagnostics-12-01775-t002:** Signal distribution pattern in the NIR-FOI of the hands of patients with rheumatic diseases.

Region	Name	Tissue	Visible in Phase	Depth of Location	Perfusion
Fingers	f, I	Capillary network in the nail bed	Phase 1	Superficial	High
	R	Sclerosed or necrotic tissue, keratinized plaques	Phases 2 and 3	Superficial	Low
Joints	D, P, M, C	Capillary network of the synovial membrane	Phases 1 and 2	Superficial/Deep	High
Vessels	V	Large veins on the back of the hand	Phases 2 and 3	Superficial	High
Connectivetissue	E, B	Tight connective tissue of dorsal entheses, tendons, aponeuroses	Phases 2 and 3	Deep	Low
	Y	Connective tissue/muscle	Phases 2 and 3	Superficial	Low/High
	e	ECU tendon	Phases 1 and 2	Superficial	Low
	a	Entheses and tendons between nailbed and DIP	Phases 2 and 3	Deep	Low
Muscle	m	Muscle tissue	Phase 3	Deep	High
Skin, connective tissue	W, F	Arteries/capillaries	Phase 1	Superficial/Deep	High

## Data Availability

Publicly available datasets were analyzed in this study. These data can be found here: (Xiralite. Diagnostics2022. Retrieved from https://osf.io/bqwa3, accessed on 30 May 2022).

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
