# Peer review of "Features Found in Indocyanine Green-Based Fluorescence Optical Imaging of Inflammatory Diseases of the Hands"

_diagnostics, 2022, doi:10.3390/diagnostics12081775_

Round 1

Reviewer 1 Report

Reviewer comments

Title

Features found in ICG-based fluorescence optical imaging of 2 inflammatory diseases of the hands

Overall, this is a clear, concise, and well-written manuscript. The introduction is relevant, and theory based. Overall, this is a high-quality manuscript that has implications; however, specific modifications are needed as the following:

Title

It is not appropriate to include abbreviations in the title

inflammatory diseases…do the author mean inflammatory arthritis as inflammatory diseases of the hands are multiple and variable and include infection

Abstract:

12

Osteoarthritis >>not an inflammatory disease, it is a degenerative disease

Abstract does not express exactly the methods included in the manuscript   

Introduction

 e.g., the author used this abbreviation multiple times before the references which is not appropriate

Materials and Methods

Line 51 ...the inclusion criteria is not mentioned

Line 52 ... from several clinical partners. Any partners or they had specific specialty

Line 56... After thorough analysis several patients…comma after analysis

Line 59. in [14]. not appropriate sentence

Line 77…and standardized (mp4) format at [18] …the authors here added preposition before the reference and this not appropriate

The ethical approval was not included

Statistical analysis was not also included

Results

The subject is an 51 y/o volunteer…. any included subjects should be mentioned first in the section of materials and methods

Line 86 ... hand) there is a…add a comma before there

Line 107 ... clinical diagnosis of osteoarthritis 107 and tendinitis. What is meant by tendinitis with osteoarthritis ..this description is the typical of dactylitis which is a characteristic feature of seronegative arthropathy

Author Response

Diagnostics 1774514

Answers to Reviewer 1:

- It is not appropriate to include abbreviations in the title
> ICG was changed to indocyanine green in the title.

- inflammatory diseases…do the author mean inflammatory arthritis as inflammatory. Diseases of the hands are multiple and variable and include infection.
> That's right. Not only arthritis was studied, but also inflammatory changes in other anatomic structures of the hand. The ability to image inflammation of such structures is also an advantage of FOI.

- Osteoarthritis >>not an inflammatory disease, it is a degenerative disease
> Osteoarthritis has been considered a prototypical non-inflammatory arthropathy because neutrophils are absent in the synovial fluid, as are systemic manifestations of inflammation However, the involvement of an inflammatory component, which is marked by symptoms such as joint pain, swelling and stiffness, is now well recognized. The inflammation in the hands examined here can be triggered by various causes, e. g. infection, autoimmunity, or degeneration. The English term osteoarthritis characterizes this disease in its name as inflammation, although these processes tend to play a role in advanced disease.

- Abstract does not express exactly the methods included in the manuscript
> The abstract has been modified to more clearly describe the methods used in the manuscript.

- e.g., the author used this abbreviation multiple times before the references which is not appropriate
> e.g., in the references was omitted, assuming that the reader does not expect a complete list of existing references

- Line 51 ...the inclusion criteria is not mentioned
> inclusion criteria were added

- Line 52 ... from several clinical partners. Any partners or they had specific specialty
> Added: … with speciality in rheumatology …

- Line 56... After thorough analysis several patients…comma after analysis> Answer: comma added

- Line 59. in [14]. not appropriate sentence
> Answer: sentence changed

- Line 77…and standardized (mp4) format at [18] …the authors here added preposition before the reference and this not appropriate
> The preposition was removed

- The ethical approval was not included
> The ethical approval has been added to the editorial office

- Statistical analysis was not also included
> A statistical analysis is not appropriate for the given data, because only single cases are presented. No causal dependency of the features to any diagnosis is claimed. After the exact characterization of the imaging patterns and assignment of the anatomical structures here, statistical analyses of all examinations can be carried out in further studies. First results were presented at this year's EULAR conference, two publications added: Kiesel et al. [21] and Stumper et al. [22]. Further investigations are in the status work in progress.

This was added to the discussion.

- The subject is a 51 y/o volunteer…. any included subjects should be mentioned first in the section of materials and methods
> All included subjects are listed in the materials and methods section.

- Line 86 ... hand) there is a…add a comma before there
> comma was added

- Line 107 ... clinical diagnosis of osteoarthritis and tendinitis. What is meant by tendinitis with osteoarthritis. This description is the typical of dactylitis which is a characteristic feature of seronegative arthropathy
> The term dactylitis is used in clinical practice for inflammation affecting the entire finger. All joints (DIP, PIP, MCP) and the connective tissue structures surrounding them are usually affected (segmental pattern). This is commonly found in patients with psoriatic arthritis. In osteoarthritis, it is mainly the connective tissue structures around the DIPs that are inflamed. The entheses and the tendons that come from them are also part of it. In this patient case, the tendinitis was also clinically diagnosed. That's why this case was chosen for the presentation here. As the disease progresses, the inflammation can also attack the joint itself. The cause can be degenerative (also as a result of trauma) - osteoarthritis - or autoimmune - e. g. rheumatoid arthritis. Treatment differs significantly and FOI can distinguish between these disorders (early signals in P1 often in MCP: RA, later signals in P2-3 in the DIP and PIP region: OA). 

Reviewer 2 Report

This paper describes the application of FOI in the examination of different kinds of rheumatology which will surely be advantageous for the the advancement of this field. For medicinal area, the conculsion is drawn based on database. In this work, 4500 samples have been collected but only 6 samples have been shown since they belong to different kinds of rheumatology. However, what's the situation for other left 4404 samples. Whether they all fit to your description or only the prominent or typical examples have been shown. Since this will be a general method for diagnostics, the results of other samples should be classified and at least described in the manuscript. Is there any mistake or unqualification between your method and other diagnostic method? What's the percentage? Wether they all fit well or the 100% fitting account for how many percentage? The sample in Figure 1 P3 Y is healthy presenting red region. For disease model in Figure 3 and 4  P3 Y is more distinct red. So what's the difference between Figure 1 P3 Y and Figure 3 and 4 P3 Y, since they all present red.

Author Response

Diagnostics 1774514

Answers to Reviewer 2:

- This paper describes the application of FOI in the examination of different kinds of rheumatology which will surely be advantageous for the advancement of this field. For medicinal area, the conclusion is drawn based on database. In this work, 4500 samples have been collected but only 6 samples have been shown since they belong to different kinds of rheumatology. However, what's the situation for other left 4404 samples. Whether they all fit to your description or only the prominent or typical examples have been shown.
> The task of this work is to characterize patterns of enhancement of ICG contrast agent in the FOI examinations of patients. Further to assign them to inflammatory anatomical structures, which are described as typical for the disease in clinical diagnosis. We have analyzed all the material in the database and there is surely some variability in the features found in the patient. But the analysis of this variability is beyond the topic of this work.

This was added to the discussion.

- As said id the materials and methods section, in this work prominent, typical examples are shown. Since this will be a general method for diagnostics, the results of other samples should be classified and at least described in the manuscript.
> The description of all examinations in the database is far beyond the scope of this work and will therefore not be part of this work. It will surely be addressed in future work. First results of other publication are added: Kiesel et al. [21] and Stumper et al. [22].

- Is there any mistake or qualification between your method and other diagnostic method? What's the percentage? Whether they all fit well or the 100% fitting account for how many percentages?
> This work does not state any diagnostic value of the demonstrated features, only their appearance in some typical cases is shown to draw attention to their possible value in the diagnostic process. Hence, a comparison to other diagnostic methods does not apply. However, other working groups have already compared the method with other imaging techniques, including statistical evaluation (references in the manuscript: 1-4,7, 11). There are comparisons with X-ray, ultrasound, and MRI.

This was added to the discussion.

- The sample in Figure 1 P3 Y is healthy presenting red region. For disease model in Figure 3 and 4 P3 Y is more distinct red. So what's the difference between Figure 1 P3 Y and Figure 3 and 4 P3 Y, since they all present red.
> Inflammatory reactions in the forearm can also occur in otherwise healthy patients as a result of degenerative processes associated with advancing age, trauma or overexertion in the context of professional activities or the pursuit of a hobby (Figure 1). These degenerative changes in the most stressed transitions from the forearm muscles to the power-transmitting tendons are found in many patients and may often be the cause of diagnostics by rheumatologists. The very late signal increase in P3 is typical of the enrichment pattern. This can be seen here in an otherwise clinically healthy subject.

This was added to the discussion.

Reviewer 3 Report

The topic is very interesting and the paper well written.

"approximately 4.500 FOI examinations "please specify the exact number, if possible detailing how many with each disease

How many patients included?

I would specify in more details the aim and the type of the study. Is it a pilot study? 

Any objectable data to be reported?

Limitations of the study must be provided

Discussion and conclusions need to be developed further

Author Response

Diagnostics 1774514

Answers to Reviewer 3:

- "Approximately 4.500 FOI examinations "please specify the exact number, if possible detailing how many with each disease
> A check of the database revealed that there are 3.690 patients in the database, of which 111 had multiple examinations (up to 6 follow-up examinations).

The numbers were concretized in the materials and methods section of the manuscript.

- … how many with each disease …? How many patients included?
> 291 with suspicion of early arthritis, 1174 with Rheumatoid arthritis, 741 with Psoriasis and with suspicion of PsA, 676 with Osteoarthritis, 148 with connective tissue diseases and the others with metabolic arthropathies, spondylarthritis and infectious arthritides.

The numbers were added in the materials and methods section of the manuscript.

- I would specify in more details the aim and the type of the study. Is it a pilot study?
> The task of this work in a form of communication is to characterize patterns of enhancement of a contrast agent in the FOI examinations of patients and to assign them to inflammatory anatomical structures, which are described as typical for the disease in clinical diagnosis. The diagnosis of rheumatic diseases is often difficult due to the overlapping of symptoms and affected structures. Therefore, we have focused here on attributing the patterns of the FOI to the anatomical structures and not discussing a diagnosis.

This was added to the introduction of the manuscript.

- Any objectable data to be reported?
> There are several issues that occur in the data. In some cases the injection of the contrast agent is dalayed or premature. In some cases patients were wearing artificial fingernails, which destroy the fluorescence signal in the finger tips. Also tattoos impaired the images in some cases.

This was added to the materials and methods section.

- Limitations of the study must be provided
> This study has clear limitations. Limitations of the study are, first, that only a small number of examinations can be shown as typical cases, and the choice of these cases is subjective. However, due to the large number of inspected examinations, the chosen cases are the result of well-trained knowledge. Second, “only” eighteen image features in FOI examinations are presented here. There are certainly more image features that may be relevant for diagnosis, maybe even features that are hard to see for the human eye, but that can be detected with the aid of computers. They may be introduced, discussed and added in future work. Third, there is surely a considerable amount of variability in the image features of the patients, also with identical clinical diagnosis.

This was added to the discussion.

- Discussion and conclusions need to be developed further
> The discussion was extended to include arguments about the healthy volnteer, possible objectable data, limitations of the study, reference to statistical evalution of parts of the data in EULAR Papers (Kiesel, Welker) with image features for differential diagnosis. Two references were added: Kiesel et al. [21] and Stumper et al. [22].  Possible future work build on the presented work were added.

A conclusion section was included in the manuscript.

Round 2

Reviewer 1 Report

Accept 

Reviewer 3 Report

The Authors made good efforts in the attempt to ameliorate their paper. It now merits publication.